# Microstructure and Mechanical Properties of PU/PLDL Sponges Intended for Grafting Injured Spinal Cord

**DOI:** 10.3390/polym12112693

**Published:** 2020-11-16

**Authors:** Anna Lis-Bartos, Dariusz Szarek, Małgorzata Krok-Borkowicz, Krzysztof Marycz, Włodzimierz Jarmundowicz, Jadwiga Laska

**Affiliations:** 1Department of Biomaterials and Composites, Faculty of Material Science and Ceramics, AGH University of Science and Technology, 30-962 Kraków, Poland; annal@agh.edu.pl (A.L.-B.); krok@agh.edu.pl (M.K.-B.); jlaska@agh.edu.pl (J.L.); 2Emergency Medicine Centre, Department of Neurosurgery, Lower Silesia Specialist Hospital of T. Marciniak, 50-367 Wrocław, Poland; 3Electron Microscopy Laboratory, Wroclaw University of Environmental and Life Sciences, 50-367 Wrocław, Poland; krzysztofmarycz@interia.pl; 4Department of Neurosurgery, Wroclaw Medical University, 50-367 Wrocław, Poland; jarmund@wo.pl

**Keywords:** polyurethane, polylactide, elastic sponges, biomaterials for nerve regeneration, spinal cord

## Abstract

Highly porous, elastic, and degradable polyurethane and polyurethane/polylactide (PU/PLDL) sponges, in various shapes and sizes, with open interconnected pores, and porosity up to 90% have been manufactured. They have been intended for gap filling in the injured spinal cord. The porosity of the sponges depended on the content of polylactide, i.e., it decreased with the increase of polylactide content. The rise of polylactide content caused an increase of Young modulus and rigidity as well as a more complex morphology of the polyurethane/polylactide blends. The mechanical properties, in vitro toxicity, and degradation in artificial cerebrospinal fluid were tested. Sponges underwent continuous degradation with varying degradation rates depending on the polymer composition. In vitro cell studies with fibroblast cultures proved the biocompatibility of the polymers. Based on the obtained results, the designed PU/PLDL sponges appeared to be promising candidates for bridging gaps within injured spinal cord in further in vitro and in vivo studies.

## 1. Introduction

Injuries of the central nervous system (CNS) are serious medical cases that usually lead to severe disability. They are especially incapacitating in case of spinal cord injuries (SCI). Regardless of the location, central nervous system injuries always cause a loss of important neurological functions [1,2,3,4,5,6,7,8,9,10]. Treatment is difficult, and at the current state of medical knowledge, it is focused on reducing the primary and secondary damages caused by the injury, and, later on, on neurological improvement by long-time rehabilitation. Full functional recovery in the case of major trauma is seldom possible to date [1,2,3,4,5,6,7,8,9,10,11,12,13,14,15].

If the injury occurs with a loss of nervous tissue, a gap is created in the spinal cord pathways, and the stumps become covered with a glial scar. It was found that axons of long neuronal tracts in the split spinal cord start to regrow in the same way as in the peripheral nervous system, but cystic gaps and glial scars block this process. In recent years, growing interest has been aimed at the use of Schwann [16] or olfactory ensheathing cells [17,18,19] to support central system nerve regeneration. These cells have unique abilities to open glial scars, making them penetrable for regrowing axons. Another goal in a successful treatment of the spinal cord injury is bridging the stumps with an appropriate material, which has to be non-toxic for the extremely sensitive nervous tissue. It should also mimic the extracellular matrix. Three-dimensional artificial scaffolds have been tested as an experimental strategy to increase axonal guidance in spinal cord lesions [5,15,16,18]. 

The idea of applying polymer scaffolds facilitating the restoration of neurological functions after spinal cord injury has been increasingly investigated. An ideal scaffold has to be penetrable for regrowing axons of the spinal cord, provide a cell-“friendly” surface for cellular colonization, provide mechanical support for regrowing nervous tissue, and connect ends of injured nervous pathways [15,16]. Biocompatible scaffolds supporting tissues have been produced with the use of natural or synthetic polymers [20,21,22]. An artificial extracellular matrix (ECM) for tissue engineering should be biocompatible without the tendency to induce exaggerated inflammatory reaction after implantation [23,24]. In order to regenerate and restore the damaged or lost tissue and organs, 3D architecture that guides the growth of new tissue must be designed with appropriate biomechanical properties for the cells adhesion and optimal integration with the host tissue [23]. In general, artificial ECM implants should exhibit a porous structure with interconnecting pores of adequate size to allow a rapid and sufficient vascularization and cell infiltration with the possibility for the long-term survival of cells within the structure [25,26,27,28]. Three-dimensional porous polymeric matrices for tissue engineering are promising constructs. These synthetic structures might offer a biological substitute for the natural extracellular matrix by directing the organization, proliferation, and differentiation of various cells and tissues [21,26,29,30]. It is widely accepted that the one of the most important factors in neuroregeneration inside an implant is a presence of longitudinal pathways guiding the growing axons [31].

Grafting the injured spinal cord with artificial implants is of great importance in the medical treatment [1,3,4,5,6,7,8,9,10]. For that purpose, polymer biomaterials have been thoroughly investigated [1,2,3,4,5,6,7]. Natural polymers, such as alginate, agarose, collagen, chitosan, and fibronectin have been used as scaffolds for neural cell growth in vitro [10] as well as axon regrowth in vivo [5]. It was found that a natural polymer may potentially increase the interaction of the host tissue with the scaffold. Nevertheless, the synthetic polymers such as poly(lactic acid), poly(glycolic acid), or poly-β-hydroxybutyrate have been tailored to create a wide range of degradation rates and mechanical properties. Finally, synthetic and natural polymers can be blended together to achieve the superior material properties [1,3,4,5,6,7,8,9,10,32,33,34,35,36,37,38,39]. A polymer scaffold for spinal cord repair should be constructed into specific geometry and micro-architecture, e.g., polymer sponges, multichannel grafts, hydrogels, and tubes [1,4,6,7]. Numerous techniques of preparation of polymer porous scaffolds are described, such as electrospinning, solvent casting/salt leaching, phase inversion, thermally induced phase separation, etc. [22,28,29,30,31,32,33,34,35,36,37,38,39,40,41,42,43,44,45,46]. One of the most common is the solvent-casting/salt-leaching technique.

In this article, the fabrication of highly porous polyurethane and polyurethane/polylactide scaffolds designed for spinal cord posttraumatic cavities reconstruction is reported. Three-dimensional sponges were made by a solvent-casting/salt-leaching technique using the dimethylformamide as a solvent. The effect of the polymer blend composition on the porous structure and morphology was examined by scanning electron microscopy. The mechanical properties of created scaffolds, porosity, and degradation in an artificial cerebrospinal fluid were studied. To test the cytotoxicity of the scaffolds, studies of the cellular response using fibroblast cultures were performed. Recently, we successfully assessed in in vitro and in vivo studies that the PU/PLDL (poly(L/D,L-lactide) blends are very promising biomaterials for peripheral and spinal cord nerve regeneration [5,39,43,47,48,49,50]. The influence of sterilization on the PU/PLDL blends was also discussed earlier [51].

## 2. Materials and Methods

Biodegradable polyurethane for biomedical applications (PU) was purchased from BAYER Material Science Company, Leverkusen, Germany. Poly(L/D,L-lactide) (PLDL), with a molar ratio L-lactide to DL-lactide 80:20% (PURASORB^®^ PLDL8038) was purchased from PURACbiochem BV, Gorinchem, Netherlands. Dimethylformamide (DMF) as a solvent was purchased from POCH S.A., Gliwice, Poland. Sodium chloride (POCH S.A.) was used as porogen.

Porous polyurethane and PU/PLDL blend sponges were prepared by the solvent-casting/salt-leaching (SCSL) method. Solutions of concentration 10 wt% were prepared by dissolving the polymer granules in DMF during 3-day stirring at 40 °C. The resulting solution was used for the pre-form preparation. Polymers solution was mixed with sieved salt grains (300–600 µm). The mixture was mechanically stirred at room temperature for 30 min. Then, the mixture was poured into 5-mL glass beakers and kept for one hour under a fume hood. Excess polymer solution was removed using micropipettes. The formed polymer solution/porogen mixture in glassware was air-dried for 7 days and then vacuum-dried for 2 days followed by salt leaching in distilled water until the electrical conductivity of the rinse water was about 2 × 10^−3^ Sm^−1^. Sponges of various compositions of PU and PLDL, namely, 90/10, 80/20, 70/30, 60/40 and 50/50 wt% were fabricated.

The cross-sectional and surface morphology of the polyurethane and PU/PLDL sponges was observed with scanning electron microscopy SEM (JOEL Model JSM-5400, NANO SEM 200 FEI EUROPE COMPANY, Eidhoven, Netherlands, and EVO LS15—Zeiss Oberkohen, Carl Zeiss, Oberkochen, Germany) at an accelerating voltage of 10–20 kV. For SEM investigation, samples in the form of discs were cut from cylindrical sponges. Polymer sponges were sputtered with carbon or gold (Turbo Dual Head Coater type K575XD and Scancoat6—Edwards, Warsaw, Poland). Samples exhibiting the cross-section surface of the sponges were fixed on sample holders using the conductive type.

Spinal cord of adult gecko Eublepharis macularius and laboratory white rat (sacrificed in other experiment) were harvested for morphological analysis, postfixed in formalin, and embedded in paraffin. Then, the samples were cut in 20 µm slices, sputtered with gold, and observed in SEM microscopy with the same technique as for PU/PLDL sponges.

Porosity evaluation of the PU and PU/PLDL sponges were calculated basing on their apparent densities and the densities of the bulk PU-based material. The weight, height, and diameter of the porous materials were delimited to calculate the volume and apparent densities. The determination of the volume and densities of the nonporous polymer samples was performed as described for porous sponges. For this purpose, finely ground material of the sponges was used. For each sample, ten specimens were tested.

In vitro degradation studies were carried out by immersing the samples in an artificial cerebrospinal fluid (ACSF, pH = 7.4) at 37 °C (MEMMERT incubator, EQUIMED, Cracow, Poland). The pH of the extracts was measured regularly during one year. The ACSF was replaced 4 times every day, as it occurs in the living organism. Glucose-free ASCF was prepared in the laboratory by dissolving appropriate salt (POCH S.A., Gliwice, Poland) in water. One L of the ASCF contained in mM: NaCl 128, KCl 3.0, CaCl_2_ 1.3, MgCl_2_ 1.0, Na_2_HPO_4_ 21.0, NaH_2_PO_4_ 1.3, and 5% CO_2_. Cylindrical samples of the polymer sponges, 10 mm in diameter and 10 mm in height, for the degradation tests were fabricated. In addition to the pH and conductivity compression strength, the Young moduli, as well as the weight loss of the sponges were determined. Samples were taken at intervals and weighed (moisture analyzer, RADWAG, Radom, Poland) after drying in a vacuum for 2 days. The remaining weight (WR) was calculated as:WR (%) = 100 × (WB − WD)/WB(1)
where WB and WD are the weights of sponges before and after immersion in ACSF, respectively. All the given numbers were means of ten measurements (±standard deviation)

A universal machine (Zwick & Rockwell 1435, Zwick/Roell, Ulm, Germany) was used for the measurement of the compression strength. Cubic (a = 10 mm) or cylindrical (d and h = 10 mm)-shaped samples were prepared for mechanical testing. The fully dried samples were compressed between parallel plates. In a first series of experiments, the compression strength (RC0.1) at the point of 10% compression (strain = 0.1) and compressive Young modulus (E) of the porous samples were measured immediately after their fabrication at ambient temperature (25 °C). As it was stated earlier in the text, to investigate the changes in mechanical properties during the early stage of degradation, sponges after immersion in ACSF were dried, and compression mechanical tests were conducted as described above. The compressive strain–stress curves of all sponge series were determined with a 10 N load cell and a cross-head speed of 1 mm/min. The compressive Young’s modulus (MPa) was calculated for the force range between 0.5 and 1 N from a stress–strain curve. All the given values were means for ten measurements (±standard deviation)

For in vitro bioassay, the discs with dimensions of 15–16 mm in diameter and 2 mm of thickness cut from sponges were sterilized using a hydrogen peroxide cold plasma sterilization technique. Then, they were located in 24-well plates (SARSDTET) and fixed in the inserts. Two types of polymer sponges were selected for biological studies: pure polyurethane and polyurethane/polylactide with weight content ratio of 8:2. The NIH 3T3 mice embryonic fibroblast cell line was seeded into the PU and PU/PLDL composite sponges by dropping the cell suspension solution onto the sponge disc. Cells were seeded on both sides of the porous disc (bottom and top surface). The initial cell density was 35,000 cells per well. The cells were cultured for 1 and 3 days on the considered sponges in Dulbecco’s Modified Eagle Medium (DMEM) supplemented with 10% phosphate buffered saline (PBS), 1% penicillin/streptomycin, and 2 mM L-glutamine at 37 °C under 5% CO_2_ atmosphere.

Mice fibroblast proliferation was measured by determining the level of reduction of the yellow tetrazolium dye MTT (3-[4-5-dimethylthiazol-2-yl]-2,5-diphenyltetrazolium bromide) to colored blue-purple insoluble formazan (calorimetric MTT bioassay). MTT solution in PBS (5 mg/mL) was added to the cells containing wells and incubated at 37 °C for 3 h. The reaction was completed by addition of dimethyl sulfoxide (DMSO) to dissolve the insoluble formazan. Then, the optical density of the blue dye was measured at a wavelength of 540 nm on Multiscan FC Microplate Photometer (Thermo Scientific, Waltham, MA, USA).

The morphology of the fibroblasts NIH 3T3 was observed by SEM and fluorescence microscopy (Zeiss Axiovert, Carl Zeiss, Oberkochen, Germany). The cultured cells were fixed in 4% paraformaldehyde (Avantor, Gliwice, Poland) for 1 h, washed in PBS, and stained with 1% acridine orange solution (1 mg cationic dye per mL) prior to microscopic analysis.

The level of protein production by cell supernatants was calculated from the absorption spectra collected in the bicinchoninic acid test (BCA assay). The BCA protein assay kit was purchased from Sigma, Saint Louis, USA. The BCA solution (reagent A) was mixed with copper(II) sulfate pentahydrate solution (4% *w*/*v* of CuSO_4_·5H_2_O), (reagent B) in the volume ration of 50:1, respectively. Then, the supernatant (10 µL) and bicinchoninic acid (200 µL) were added to a 96-well plate. After 30 min incubation in the dark, the absorbance was determined at a wavelength of 540 nm with a photometer Multiscan FC Microplate Photometer (Thermo Scientific, Thermo Scientific, Waltham, MA, USA).

The level of nitrate(II)/nitrate(III) production, as an indicator of nitric oxide (NO) synthesis, was quantified in fibroblast culture supernatants by the Griess diazotization reaction (nitric oxide calorimetric bioassay). Griess reagents for nitrite determination were purchased from Sigma, Saint Louis, USA To perform this bioassay, a Griess component A–0.1% (1 mg/mL) N-(1-naphtyl)ethylenediamine dihydrochloride solution in water and Griess component B–1% sulfanilic acid in 5% phosphoric acid solution (component C) were mixed in the volume proportion 1:1. Then, 100 µL of the supernatant and 100 µL of the Griess reagents (A and B-C) were transferred to a multi-well plate (96 wells). The absorbance was measured at wavelength of 540 nm with a photometer Multiscan FC Microplate Photometer (Thermo Scientific, Waltham, MA, USA).

The results were expressed as means ± standard deviation. Statistical significance was determined by *t*-test and the differences were regarded as significant at *p* < 0.05.

## 3. Results and Discussion

In our work, we designed and fabricated the PU-based sponges that may act as grafts, providing support and contact guidance for regrowing axons and endogenous and/or transplanted cells. On the basis of our earlier research (unpublished data), we found that a material/scaffold suitable for spinal cord reconstruction should at least fulfill the following basic criteria: (1) biodegradable into non-toxic products, (2) high elasticity, (3) high resistance to compression, (4) compression modulus about 1–1.5 MPa [52,53], (5) interconnected pore structure mimicking the spinal cord architecture, (6) micropores ranging in size from 100 µm to 1 mm [10,15,18], and (7) be easily fabricated into a variety geometries and dimensions [15,16]. Its design was drawn from SEM and classic histological and anatomical studies of spinal cord architecture. It led us to focusing on the sponge material as a matrix for implants. It was based on their resemblance to the main microscopic architecture of spinal cord and 3D matrix formed by glial supportive cells, as shown in Figure 1.

### 3.1. SEM Studies, Porosity

Various forms of polymer sponges consisting of polyurethane and polylactide prepared by the solvent-casting/salt-leaching method are shown in Figure 2. The preparation method showed very good reproducibility. The samples were fabricated in different shapes appropriate for the mechanical tests. As we can see on the SEM images of rat and leopard gecko spinal cord cross-sections in the thoracic level (Figure 1), they have porous architecture. The porosity of both kinds of spinal cord samples is not uniform. The white matter that consists mainly of axons and glial cells is much more porous than the gray one containing numerous cell bodies. It is especially noticeable in the rat spinal cord. One can expect that porosity should be a crucial property of material grafting the disrupted spinal cord.

Figure 3 shows SEM micrographs of the cross-sections (images b–f) of the PU/PLDL sponges. All samples showed a highly microporous structure. The pores in the sponges are interconnected and regularly distributed. Mostly large micropores with the diameter of >300 µm are present; however, the diameter of pores varied between 50 and 600 µm. In Figure 3a,g SEM images of sole PU and PLDL sponges are presented. There were no significant differences in the porosity of the sponges made of pure polyurethane and the PU/PLDL blends. On the contrary, pure polylactide samples obtained with exactly the same method as the PU and PU/PLDL ones showed pores with diameters less than 10 µm. The diameters of pores in rat and gecko spinal cords range from the nanoscale up to 10 µm. Even if the PLDL sponge has better resemblance to the native nerve, they are too hard and rigid for grafting the spinal cord. The dimensions of the pores in PU/PLDL correlate well with the size of the porogen and can be modified this way. Figure 3h,i shows two representative SEM micrographs of the external surface of PU/PLDL 8/2 sponges taken with two different magnifications. Open pores are clearly visible, which potentially will let the body fluids and cells inside the material.

The porosity of polyurethane and PU/PLDL sponges is presented in Figure 4. The estimation of porosity gave values between 75 and 85% for all samples, independent of the initial concentration of the solution. However PU/PLDL 8/2 sponges have porosity in the higher range, which is more than 80%. Depending on the polymer contents, the morphology (mainly size of the pores) of the samples slightly differs (Figure 3); however, no clear rule can be seen. There were significant differences in the porosity and morphology of pure polyurethane sponges and pure polylactide sponges, which is shown in Figure 3a,g.

The porosity of the manufactured sponges was determined by measuring the apparent density of the sponges and the densities of the corresponding solid polymeric materials. Based on these densities, the porosities of the scaffolds were calculated to range between 70% for PU/PLDL 5/5 composite and above 85% for PU/PLDL 8/2 composite porous scaffolds (Figure 4). Summarizing, the morphology and porosity of the PU/PLDL sponges can be easily controlled by the amount of sodium chloride added to the polymer solution and the particle size of the porogen.

### 3.2. Degradability of the Sponges in the Artificial Cerebrospinal Fluid

The in vitro degradation of polyurethane, polylactide, and PU/PLDL sponges was performed in artificial cerebrospinal fluid (ACSF) at 37 °C by measuring the weight loss and mechanical properties alterations of the scaffolds as a function of the incubation time. The total volume of CSF in an adult human body ranges from 140 to 270 mL. CSF is produced at a rate of 0.2 ± 0.7 mL per minute (about 500 ÷ 700 mL per day), so the whole fluid volume changes every 6 to 7 h [54,55,56,57,58]. In our study, we kept replacing the ACSF four times a day during whole period of the experiment. In Figure 5, changes of mass loss upon degradation in ACSF for 56 weeks are presented.

The immersion of all sponges in artificial cerebrospinal fluid resulted in a gradual decrease of mass. The tests have been performed for 56 weeks. Polyurethane and polylactide were also tested. Their degradation rate differs significantly. The PLDL sponges degrade completely in 28 weeks, while polyurethane ones show only 30% mass loss after 56 weeks. It is reasonable to add that the polyurethane used for the preparation of scaffolds degrades entirely in ca. 3 years. All investigated blends of the two polymers, except for PU/PLDL 5/5, last for 56 weeks; however they degrade faster than PU, and the degradation level depends on the polymer content ratio. The more polylactide, the faster the degradation. The samples containing 50 wt% of PLDL lost almost 50% of their weight after 8 weeks; similarly, the mass loss in the other samples is congruent with the content of PLDL. PU/PLDL sponges containing of more than 20 wt% of PLDL revealed a higher degradation rate during the first 8 weeks than PU and PU/PLDL 9/1. After 8 weeks, the degradation slows down. However, we can expect that the degradation mechanism in the polymer mixtures is different than in the neat polymers, and the degradation of polyurethane is faster in the presence of polylactide. The evidence for that is that the blend 5/5 degrades almost totally after 56 weeks, while sole PU does not. The PLDL sponges weight decreased progressively from 100% down to about 50% of the initial weight during 8 weeks of degradation. On the contrary, insignificant mass loss was observed for the polyurethane sponges during the first weeks.

The rate of the biodegradation of the polymer scaffolds intended for spinal cord repair must correlate with the nervous tissue regeneration rate [18]. Interpretation of the in vitro degradation results is difficult due to lack of data of the regeneration rate in the CNS on the basis of implants. For a peripheral nervous system, the rate of fiber regeneration is defined more precisely, and it is assumed to be about 1 mm/day [59,60,61,62,63].

### 3.3. Mechanical Properties

The material grafting of the injured spinal cord should be soft and elastic to be mechanically compatible with the spinal cord tissue. On the other hand, the compressive modulus and compressive strength of the graft must be very accurate to prevent it from deformation blocking the regeneration of the nerve fibers. Stress–strain curves of PU and PU/PLDL sponges are presented in Figure 6a. All samples show viscoelastic behavior that is typical to PU. The addition of PLDL, which is rigid in its nature, changes the character of the blends form nonlinear viscoelastic (for 9/1 and 8/2) to almost linear elastic for PU/PLDL 5/5. In Figure 6b, the analogical compression curves of samples after 4-week immersion in ACF are shown. They are consistent with the degradation graphs in Figure 5. During the first weeks, mainly PLDL is removed from the blend, causing an increase of elasticity of the samples, which is evidenced by changing the stress–strain curves to nonlinear and is closer to that of the PU sponge. The stress–strain curve of the PU sponge remained almost unchanged after 4 weeks of treatment with ACF.

The compressive moduli (Ec) and compressive strength (Rc 0.1) at 10% deformation are summarized in Table 1. The values of the compressive moduli of all blends are typical for elastomers (usu. ≈ 1 MPa for rubbers) and also compatible with the values published for the human spinal cord [53]. The compressive moduli of the PU/PLDL sponges are visibly higher than those of the PU sponges. The bigger the PLDL contents, the higher the Rc0.1 and EC of the PU/PLDL sponges, meaning that the sponges are stiffer and more resistant to mechanical stress. The compression moduli of the sponges ranged from 0.88 ± 0.04 MPa for neat polyurethane to about 3.42 MPa for PU/PLDL 5/5. It refers to human spinal cord Young moduli described by other that grows with sample size 0.025 mm-0.48 mm-0.12 mm to 0.37, 1.02, 1.17 MPa respectively [52] and is increased when covered by pia matter—1.40 MPa [53]. 

Figure 7 shows the changes of the compression moduli of the sponges after 36 weeks of incubation in ACSF. Before immersion, the samples considerably differed with their mechanical properties, and the compression moduli ranged from 0.88 to 3.42 MPa. After 4 weeks, the immersion mechanical properties of all samples became comparable to those of PU sponge. After 12 weeks degradation in ACSF, the compression moduli ranged from 0.44 MPa for PU/PLDL 5/5 to 0.73 MPa for PU. After 24 weeks, the compressive moduli of all blends were lower than of those of PU, which is in agreement with the significantly greater mass loss in blends than in PU (see Figure 5), and consequently, there are more voids in the material.

### 3.4. In Vitro Bioassay

The proliferation of fibroblasts on the tissue-culture polystyrene plate (TCPS, the control) and two types of sponges (PU and PU/PLDL 8/2 sponges) was investigated using MMT bioassay, as well as non-cytotoxic effect of PU-based sponges after day 1 and 3. The PU and PU/PLDL 8/2 sponges were selected because of their excellent elasticity, mechanical properties, and preferable morphology and degradation rate. The results of fibroblast proliferation observation in MMT assay are presented in Figure 8.

The absorbance of each type of PU-based sponge was lower to that of the control at every time point. The absorbance of both types of sponges generally increased from day 1 to day 3 but did not change significantly. On day 3, the cell proliferation was lower in porous samples, reaching about 30,000 compared to cell viability in control TCPS samples, around 320,000.

In the in vitro experiment, fibroblasts were not found to adhere to neither to PU sponges nor PU/PLDL 8/2 sponges. Figure 9 shows the representative morphology of a small singular fibroblast seeded in the pores of PU-based sponges on day 3. The cells had already attached onto the sponges and exhibited a globular shape.

To assess the ability of PU-based sponges to promote fibroblast metabolic activity, we analyzed the production of proteins and nitric oxide after 1 and 3 days. The production of protein slowly decreased from the first to the third day of cell cultivation. The production of protein increased only in the presence of the bottom side of polyurethane sponges and TCPS, decreased in the presence of the PU/PLDL 8/2 composite sponges, was slightly higher in the presence of the top surface, and showed the most significant increase in the presence of PU/PLDL 8/2 during the first day of cell culture (Figure 10).

The study of the interaction between PU/PLDL sponges and mice fibroblast-mediated nitric oxide in sponges of polyurethane and PU/PLDL 8/2 composition showed increased an production of NO in the presence of a polylactide phase of PU-based sponges and the presence of the bottom surface of polyurethane sponges, suggesting that both porous matrices stimulate nitric oxide production, although, for the top surface of PU sponges, the production of NO by fibroblast decreased (Figure 11). In summary, the protein and nitric oxide expression by fibroblast seeded on these porous scaffolds was comparable to that on the control on day 1 and day 3.

### 3.5. Summary

We have found that all PU/PLDL sponges exhibited good mechanical properties with a sufficiently low compression modulus about 1–1.5 MPa, which is important for an atraumatic contact with spinal cord lesions. Thus, they may be used in neural tissue engineering to generate tissue substitutes that will not induce a strong irritate reaction after implantation into the soft tissue of the spinal cord, and at the same time, it will also set up a stable construct that is in contact with the hard vertebral column surface. The in vitro degradation studies in ACSF revealed a continuous weight loss of all the tested samples over a degradation period of a year. The rate of degradation depended on the PLDL concentration. Young moduli of the investigated PU/PLDL sponges depended on the polymer content ratio and ranged from 0.88 MPa for PU to 1.5 MPa for PU/PLDL 5/5, and the values are similar to that of human spinal cord (1.4 MPa with pia matter, 1 MPa without pia matter). Polyurethane and PU/PLDL 9/1 sponges exhibited good mechanical properties, but the degradation rate was rather low. The initial compression modulus of 3.42 of the PU/PLDL 5/5 sponges is sufficiently high to serve as a reconstruction material. The PU/PLDL 8/2 sponges best fulfilled all the criteria to serve as a reconstruction material.

Mice fibroblast were seeded on the PU and PU/PLDL 8/2 sponges. SEM images showed they were poorly overgrown by the fibroblast. The sponges did not exhibit a cytotoxic effect on the fibroblasts, and the MMT data indicated that both types of sponges, PU and PU/PLDL 8/2, supported the expression of protein and nitric oxide throughout the fibroblast culture period of 3 days. The comparable metabolic activity of fibroblast seeded on the sponges, TCPS, and medium on day 3 suggest that the presence of PU and PU/PLDL 8/2 sponges was particularly useful for promoting the cells for the expression of protein and nitric oxide. These preliminary in vitro studies demonstrated the cytocompatibility of the fabricated scaffolds and their potential to serve as biodegradable matrices for cell culture and tissue repair. The protein and nitric oxide production by fibroblast seeded on the sponges was comparable to those of the control sample for each measurement. It is worthwhile to add that nitric oxide is a biomolecular mediator of many physiological processes, e.g., immunity, inflammation, thrombosis, and neurotransmission. In addition, our earlier detailed studies on the culture of olfactory ensheating glial cells on polyurethane/polylactide electrospun nonwovens proved the usefulness of the PU/PLDL blends for neural regeneration.

## 4. Conclusions

Applying novel techniques of tissue engineering for neural regeneration requires biocompatible and preferably bioresorbable scaffolds providing a framework for cells to adhere, proliferate, differentiate, and create an artificial extracellular matrix. Polyurethane/polylactide sponges mimicking the spinal cord tissue of rat and leopard gecko were successfully designed and prepared. Our study demonstrates that the biodegradable polyurethane/polylactide blend sponges exhibit a highly porous microstructure, good mechanical properties, and also cytocompatibility, and thus, they can be used for further design as a neural tissue ECM substitute, which can be implanted directly at the defect site in the spinal cord without inducing a strong irritation response. However, because of the unsatisfactory pore interconnectivity and the poor infiltration of body fluids in the PU-based sponges, further studies will be performed to improve the pore interconnectivity, the infiltration, and the cell adhesion, which are an essential to accomplishing optimal neural regeneration. Prepared characteristics of Pu/PLDL sponges can be used to determine the expected behavior of potential implants. The porosity, degradation time, and mechanical properties can be selected. Prepared PU/PLDL sponges may be used for spinal cord reconstruction. Future studies involving the optimization of glial, neural, and stem cells and tissue growth within the sponges and in vivo of the scaffolds materials are in progress.

## Figures and Tables

**Figure 1 polymers-12-02693-f001:**
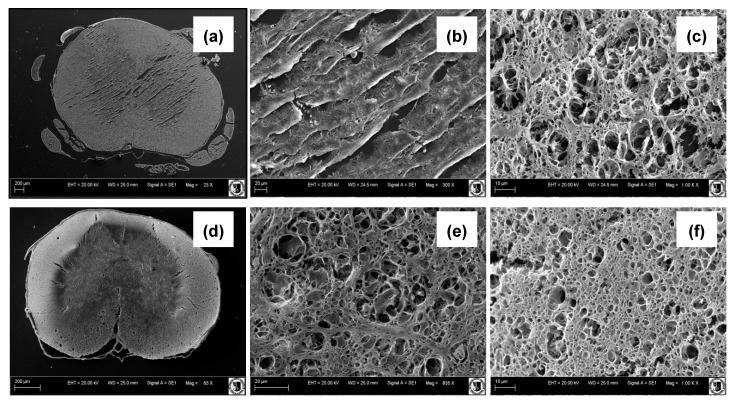
SEM images of rat (**a**–**c**) and leopard gecko spinal cord (**d**–**f**) at the thoracic level: general view (**a**,**d**), gray matter (**b**,**e**) and white matter (**c**,**f**).

**Figure 2 polymers-12-02693-f002:**
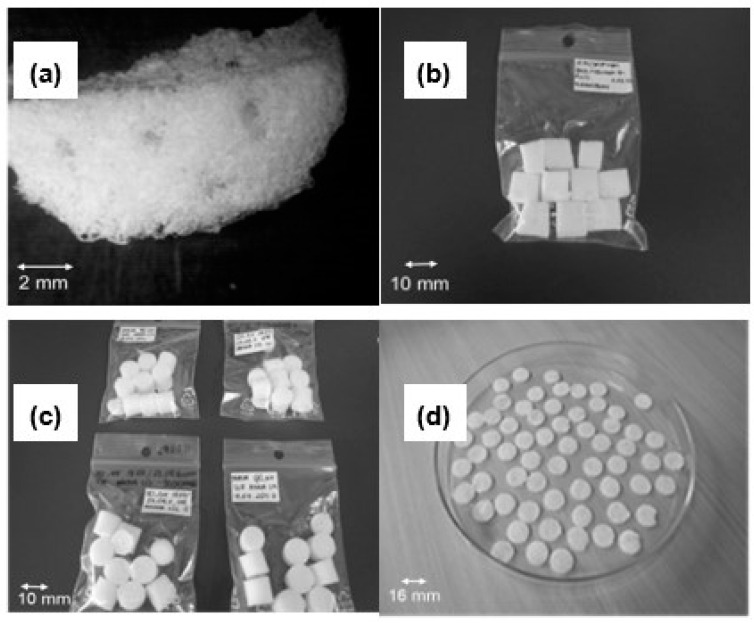
PU/PLDL (poly(L/D,L-lactide) sponges prepared by the solvent-casting/salt-leaching method: multichannel sponges (**a**), sponges for mechanical tests (**b**,**c**), samples for degradation tests (**b**), and sponges for in vitro biological tests (**d**). Scale bar 10 mm.

**Figure 3 polymers-12-02693-f003:**
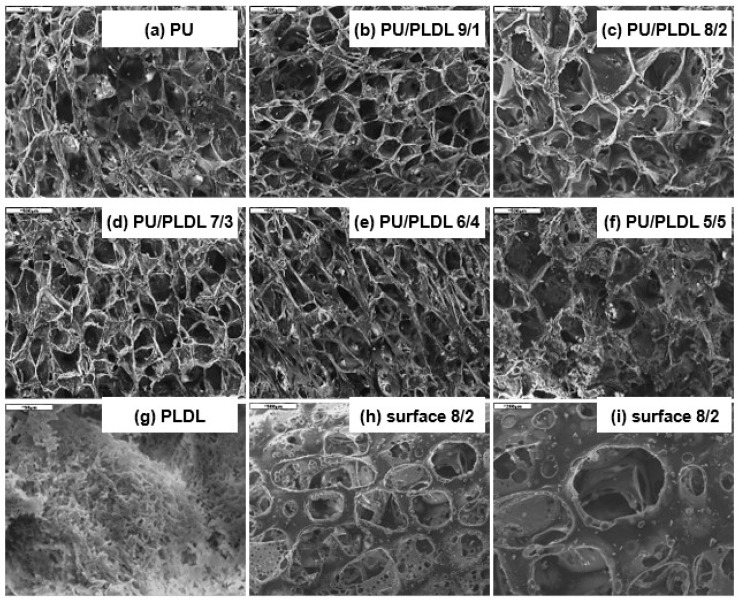
SEM images of polyurethane, polylactide, and polyurethane/polylactide sponges: (**a**–**g**) cross-sections; (**h**,**i**) surface. A composition of the specific sample is indicated in the photo.

**Figure 4 polymers-12-02693-f004:**
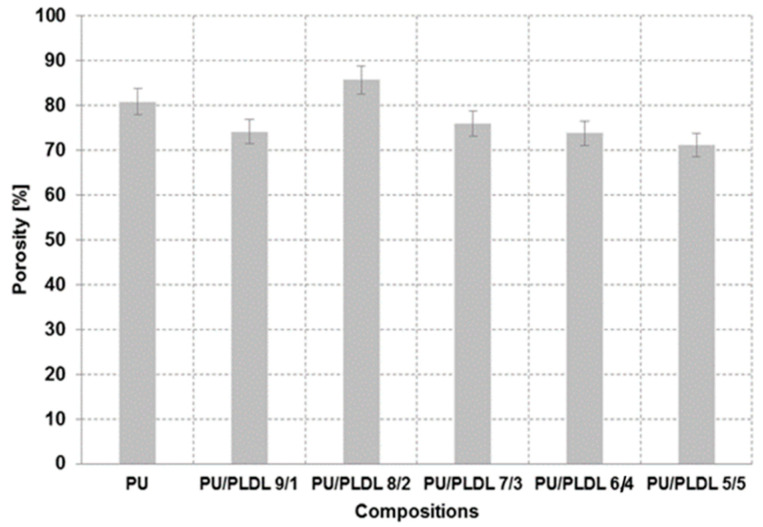
Porosity of polyurethane and polyurethane/polylactide composite sponges.

**Figure 5 polymers-12-02693-f005:**
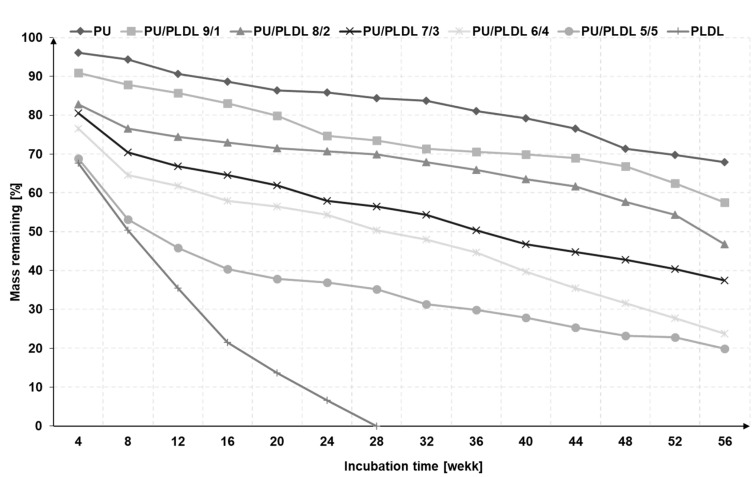
Weight loss of polyurethane, polylactide and PU/PLDL sponges as a function of degradation time in artificial cerebrospinal fluid (ACSF).

**Figure 6 polymers-12-02693-f006:**
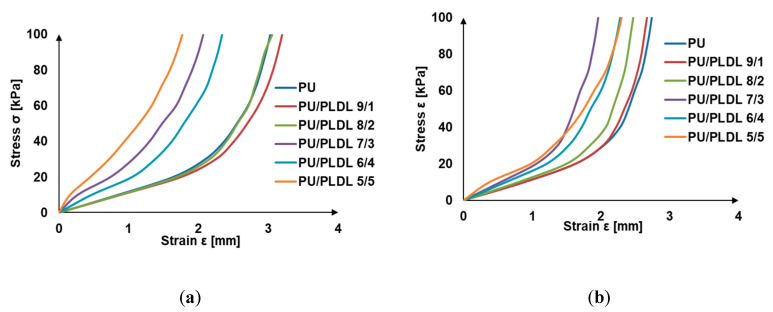
Compressive strain–stress curves of the PU/PLDL sponges: (**a**) before degradation in ACSF; (**b**) after one month incubation in ACSF**.**

**Figure 7 polymers-12-02693-f007:**
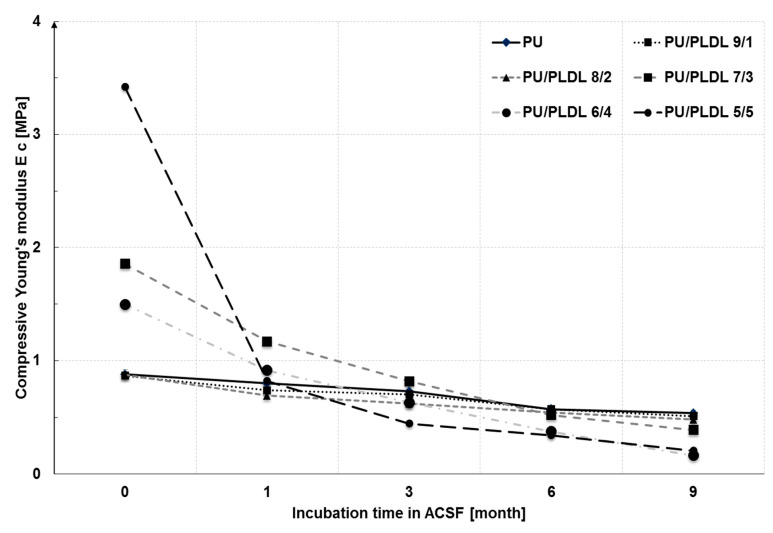
Compressive moduli of PU/PLDL sponges as a function of the degradation time in artificial cerebrospinal fluid.

**Figure 8 polymers-12-02693-f008:**
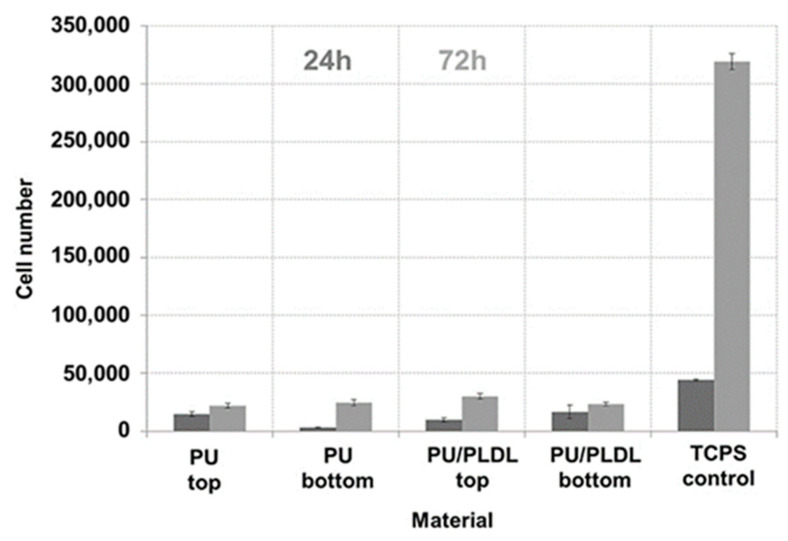
Effect of PU and PU/PLDL 8/2 sponges on the viability and proliferation of mice fibroblast NIH 3T3.

**Figure 9 polymers-12-02693-f009:**
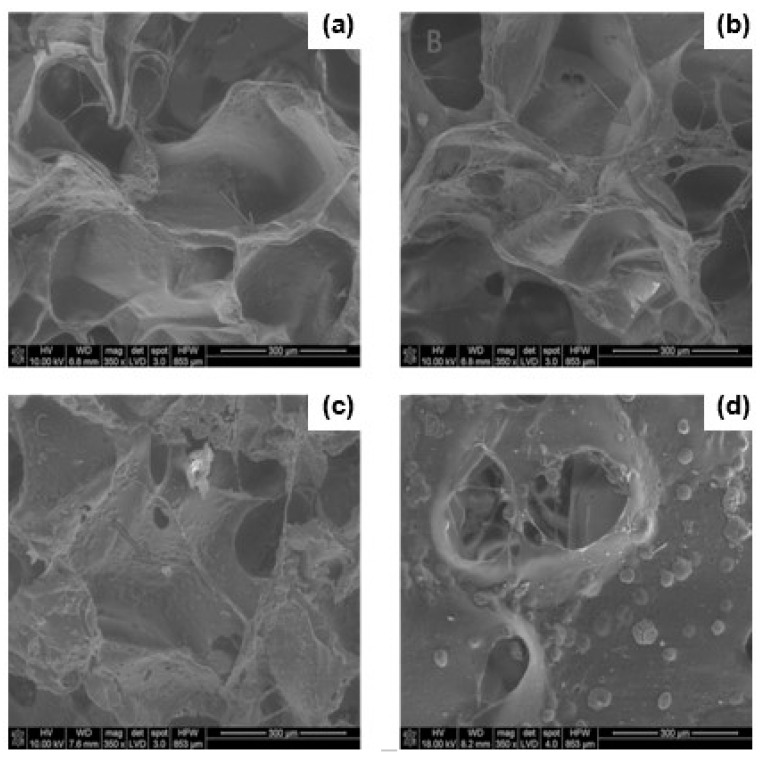
SEM micrographs of fibroblasts NIH 3T3 on the polyurethane (**a**,**c**), and polyurethane/polylactide 8/2 (**b**,**d**) sponges (350×) after 3 days of cell culture.

**Figure 10 polymers-12-02693-f010:**
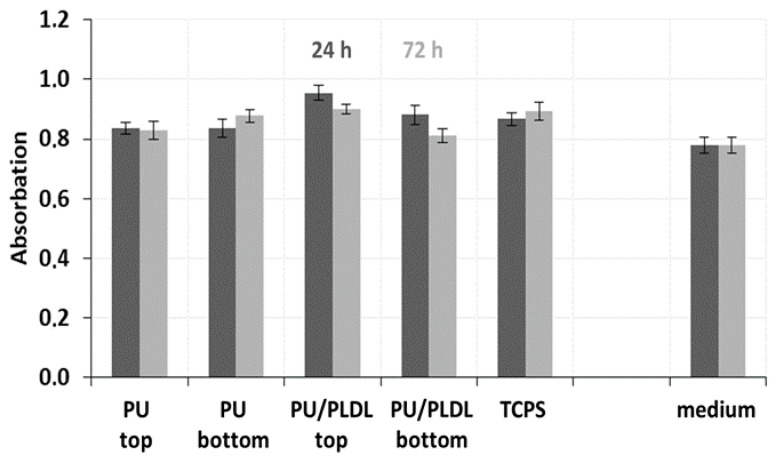
Values of proteins secretion by mice fibroblast NIH 3T3 separately for the first and third day of cell culture in the PU and PU/PLDL 8/2 sponges.

**Figure 11 polymers-12-02693-f011:**
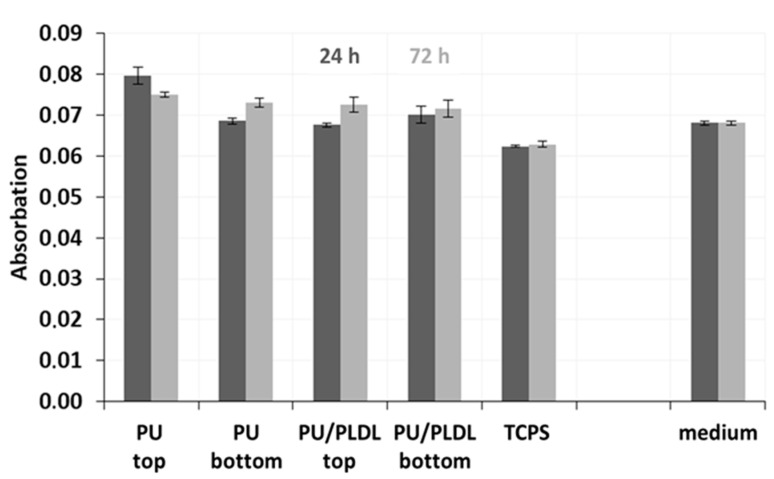
Nitric oxide (NO) production by mice fibroblast NIH 3T3 during the first and third day of cell culture in the PU and PU/PLDL 8/2 sponges.

**Table 1 polymers-12-02693-t001:** The comparison of Young’s moduli (EC) of PU/PLDL sponges measured by static compression test and compressive yield strength RC0.1 for polymer sponges.

Sponge Symbol	PU 100%	PU/PLDL 9/1	PU/PLDL 8/2	PU/PLDL 7/3	PU/PLDL 6/4	PU/PLDL 5/5
Compressive strength R_C0.1_ [kPa]	10.05 ± 0.35	10.61 ± 0.81	11.84 ± 0.76	18.20 ± 1.57	17.76 ± 2.21	26.93 ± 3.01
Young’s moduli [MPa]	0.88 ± 0.04	0.87 ± 0.05	0.85 ± 0.03	1.86 ± 0.41	1.50 ± 0.13	3.42 ± 0.23

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
