# Peer review of "Microstructure and Mechanical Properties of PU/PLDL Sponges Intended for Grafting Injured Spinal Cord"

_polymers, 2020, doi:10.3390/polym12112693_

Round 1
Reviewer 1 Report
The revision of the manuscript is attached

Author Response
Comments from Reviewer 1
Comment 1: The introduction is initially strongly focused on aspects that would be outside the current framework of the manuscript. The authors carry out a thorough review and presentation of the effects associated with spinal cord injuries, etc. This undesirably lengthens the introduction of the central focus of the work, based precisely on the synthesis of polymeric foams to promote cell growth and proliferation. Personally, the introduction actually starts on line 59 on the second page.
Response: Thank you for pointing this out. We shortened the first part of Introduction by half.
Comment 2: The authors establish the following sentence that seems confusing: “Another problem in a successful treating of the spinal cord injury is bridging the stumps with an appropriate material, which has to be non-toxic for the extremely sensitive nervous tissue”. The term "Problem" is not the most appropriate to describe a satisfactory treatment. Probably the most convenient would be to use the term "achievement" or “goal“.
Response: We agree with this and have incorporated your suggestion (goal) in the
manuscript.
Comment 3: The authors indicate: "Two main material types have been investigated for the use in neural tissue engineering: polymers [1-7], and carbon materials" however, they only make a description of the polymeric systems (natural and synthetic). The authors should briefly describe what types of carbon materials exist or rewrite the phrase indicating that polymers are the most commonly used materials for tissue engineering.
Response: Not to extend the Introduction we changed the sentence for “For that purpose, polymer biomaterials have been thoroughly investigated [1-7]”.
Comment 4: In the part of the characterization of the foams, the authors also establish several contradictory aspects: “The estimation of porosity gave values between 75 to 85 % for all samples, independent on the initial concentration of the solution.” “The pore size increased with increasing amounts of polyurethane.” “Summarizing, the morphology and porosity of the PU/PLDL sponges can be easily controlled by the amount of sodium chloride added to the polymer solution, and the particle size of the porogen.” The authors could clarify these sentences further and avoid misinterpretation.
Response: We agree with this comment, therefore in the revised manuscript the changed text can be found: “The porosity of polyurethane and PU/PLDL sponges is presented in Figure 4. The estimation of porosity gave values between 75 to 85 % for all samples, independent on the initial concentration of the solution. However PU/PLDL 8/2 sponges have porosity in the higher range, which is more than 80%. Depending on the polymer contents morphology (mainly size of the pores) of the samples slightly differs (Fig. 3), however no clear rule can be seen. There were significant differences in porosity and morphology of pure polyurethane sponges and pure polylactide sponges, what is shown in Fig. 3a and 3g.”
Comment 5: The authors state: “The dimensions of the pores in PU / PLDL correlate well with the size of the blowing agent and can be modified this way.” What do you mean by this way of modification? They should explain it better, to demonstrate the versatility of their systems, since the pore size, in principle, does not have a relationship with the composition of the blends.
Response: Yes, the pore size do not depend on the composition of the blends, however, we were able to change the size of the pores by changing size of the porogen grains. We have not describe this research in the manuscript. Not to mislead the readers we changed the sentence indicated by the Reviewer for: “The dimensions of the pores in PU / PLDL correlate well with the size of the porogen grains and can be modified this way.”
Comment 6: Regarding the mechanical properties of the foams, taking into account that the pore sizes of all studied foams are similar. why does “sample 6/4” not follow the trend shown by the rest of the foams? In other words, an increase in the content of PLA within the blend implies an increase in the modulus and compressive strength. Could it be due to an incompatibility between both polymers?.
Response: In all our studies sample 6/4 behaved a bit differently than the other ones, out of the rule. The two polymers in the blends are incompatible, but we clearly see some rules when changing the contents from 9/1 to 1/9 Pu/PLDL, only 6/4 blend seems to be “special”. We don’t have the explanation on that. For our purposes the 8/2 blend was the most suitable, than we did not focus on the 6/4. It might be, though, an interesting point to study the molecular interactions between the two polymers in the blend.
Comment 7: Figure 7 shows the variations of the foam modules after 9 months. This figure is difficult to read and interpret, in addition the indicated results were already predictable after having previously indicated the degradation rates of the foams. It does not add additional value to the quality of the article.
Response: Thank you for this comment. We think, however, that the graph adds well to the results presented in Fig. 5. We can see from Fig. 5 that the mass loss in all samples is almost linear during the 56 weeks, while the compressive moduli decrease significantly during first 4 weeks of degradation, and then they are almost comparable for all the samples. It is important to know how the mechanical properties of an implant placed in the injured spinal cord change in time. We hope that you find this explanation reasonable.
Comment 8: In the study of the proliferation of fibroblasts within the foams (section “In vitro bioassay”) the authors establish Polystyrene plates as reference. Despite being a widely used reference for this type of study, it is not the most appropriate in this case for several reasons. First, the morphology of the reference is flat instead of a foam and it is also a polymeric system chemically very different from those studied. In order to bring a high interest to the article and improve its quality, the reference should be the same polymeric system since fortunately the authors already have previous results of fibroblast proliferation for the same polymer blends (PU / PLA) but in a nonwoven arrangement. The use of these systems as a new reference could draw important conclusions on how the arrangement of the blends favours or not the proliferation of fibroblasts.
The references are:
1) Polyurethane/Polylactide-Based Electrospun Nonwovens as Carriers for Human Adipose-Derived Stromal Stem Cells and Chondrogenic Progenitor Cells. POLYMER-PLASTICS TECHNOLOGY AND ENGINEERING 2016, VOL. 55, NO. 18, 1897–1907
2) Polyurethane/polylactide-based biomaterials combined with ratolfactory bulb-derived glial cells and adipose-derived mesenchymalstromal cells for neural regenerative medicine applications. Materials Science and Engineering C 52 (2015) 163–170
3) Alternatively, other types of scaffolds using the same composition were also found in the literature and could serve as support for the writing of the results. “Characterization of thermoplastic polyurethane/polylactic acid(TPU/PLA) tissue engineering scaffolds fabricated by microcellularinjection molding. Materials Science and Engineering C 33 (2013) 4767–4776”
Response: Thank you for this suggestion. It would have been interesting to investigate this aspect. However, we plan to do that in near future as a separate study. We need to produce new nonwovens, which is time consuming. Moreover in our recent publications we proved biocompatibility of the material with cells of nervous system.
Comment 9: The solvent name Dimethylformamide is written incorrectly.
Response: Thank you for pointing this out. We have corrected all misspellings.
Comment 10: The composition 60/40 is not indicated in the experimental part.
Response: Thank you. We have corrected that.
We look forward to hearing from you in due time regarding our submission and to respond to any further questions and comments you may have.
Sincerely,
Dariusz Szarek
Reviewer 2 Report
The authors describe preparation of porous material based on the polyurethane and polylactide composition as a potential spinal cord support material. The authors have examined the toxicity, mechanical properties, and the degradation behavior. The overall results are promising and the material appears to be of interest for future studies. The technical quality of the manuscript is good. I do recommend acceptance with minor revisions.
- My major concern is the mistakes in writing throughout this manuscript. For instance, dimethylformamide is misspelled in a different ways (lines 93 and 105). Please check the whole manuscript for misspellings. Please be consistent: either use ‘%wt.’ (line 108) or ‘wt.%’ (line 116). Line 180: ‘cell ‘supernatants’ should be ‘cell supernatants’, I think. Due to inconsistencies in the writing style and the spelling mistakes, the readability of the manuscript is low. I strongly suggest that the authors carefully go through each sentence of the manuscript and revise.
- The use of term ‘nerve regeneration’ in the key words makes one believe that this is part of the experimental work. However, this is not correct.
- Figure 6: the data seems to be cut-off at 100 kPa limit. Why not show the entire data set (until the break point)?
- Figures 10 and 11: the axis legend should be ‘absorbance’ and not ‘absorbation’.
Author Response
Comments from Reviewer 2
Comment 1: My major concern is the mistakes in writing throughout this manuscript. For instance, dimethylformamide is misspelled in a different ways (lines 93 and 105). Please check the whole manuscript for misspellings. Please be consistent: either use ‘%wt.’ (line 108) or ‘wt.%’ (line 116). Line 180: ‘cell ‘supernatants’ should be ‘cell supernatants’, I think. Due to inconsistencies in the writing style and the spelling mistakes, the readability of the manuscript is low. I strongly suggest that the authors carefully go through each sentence of the manuscript and revise.
Response: Agree. We corrected all misspellings. The manuscript have been carefully checked for spelling and corrected
Comment 2: The use of term ‘nerve regeneration’ in the key words makes one believe that this is part of the experimental work. However, this is not correct.
Response: We agree with this and have added the term “biomaterials for nerve regeneration”.
Comment 3: Figure 6: the data seems to be cut-off at 100 kPa limit. Why not show the entire data set (until the break point)?
Response: Due to their high elasticity the sponges did not break under the compression stress. After the sponges were compressed to the maximum, the machine automatically stopped the test. After taking away the compressive force, the sponges returned to their primary shape.
Comment 4: Figures 10 and 11: the axis legend should be ‘absorbance’ and not ‘absorbation’.
Response: Absolutely! The axis legends have been corrected.
Round 2
Reviewer 1 Report
After reading the responses provided by the authors, I consider that they have been addressed correctly.
However, I think it is convenient to highlight that the particular behavior of the blend (60/40) should be reflected in the manuscript, since there is still no interpretation that describes this singular behavior.
In addition, I invite the authors to carry out a study focused precisely on that mixture in order to clarify this disparate behavior. This study could open new future avenues for your research.